# Robust Learning Rate Selection for Stochastic Optimization via Splitting Diagnostic

## Abstract

This paper proposes SplitSGD, a new dynamic learning rate schedule for stochastic optimization. This method decreases the learning rate for better adaptation to the local geometry of the objective function whenever a *stationary* phase is detected, that is, the iterates are likely to bounce at around a vicinity of a local minimum. The detection is performed by splitting the single thread into two and using the inner product of the gradients from the two threads as a measure of stationarity. Owing to this simple yet provably valid stationarity detection, SplitSGD is easy-to-implement and essentially does not incur additional computational cost than standard SGD. Through a series of extensive experiments, we show that this method is appropriate for both convex problems and training (non-convex) neural networks, with performance compared favorably to other stochastic optimization methods. Importantly, this method is observed to be very robust with a set of default parameters for a wide range of problems and, moreover, yields better generalization performance than other adaptive gradient methods such as Adam.

## 1 Introduction

Many machine learning problems boil down to finding a minimizer $\theta^* \in \mathbb{R}^d$ of a risk function taking the form

$$F(\theta) = \mathbb{E}\left[f(\theta, Z)\right], \tag{1}$$

where $f$ denotes a loss function, $\theta$ is the model parameter, and the *random* data point $Z = (X, y)$ contains a feature vector $X$ and its label $y$. In the case of a finite population, for example, this problem is reduced to the empirical minimization problem. The touchstone method for minimizing (1) is stochastic gradient descent (SGD). Starting from an initial point $\theta_0$, SGD updates the iterates according to

$$\theta_{t+1} = \theta_t - \eta_t \cdot g(\theta_t, Z_{t+1}) \tag{2}$$

for $t \geq 0$, where $\eta_t$ is the learning rate, $\{Z_t\}_{t=1}^\infty$ are i.i.d. copies of $Z$ and $g(\theta, Z)$ is the (sub-) gradient of $f(\theta, Z)$ with respect to $\theta$. The noisy gradient $g(\theta, Z)$ is an unbiased estimate for the true gradient $\nabla F(\theta)$ in the sense that $\mathbb{E}\left[g(\theta, Z)\right] = \nabla F(\theta)$ for any $\theta$.

The convergence rate of SGD crucially depends on the *learning rate*—often recognized as "the single most important hyper-parameter" in training deep neural networks (Bengio, 2012)—and, accordingly, there is a vast literature on how to *decrease* this fundamental tuning parameter for improved convergence performance. In the pioneering work of Robbins and Monro (1951), the learning rate $\eta_t$ is set to $O(1/t)$ for convex objectives. Later, it was recognized that a slowly decreasing learning rate in conjunction with iterate averaging leads to a faster rate of convergence for strongly convex and smooth objectives (Ruppert, 1988; Polyak and Juditsky, 1992). More recently, extensive effort has been devoted to incorporating preconditioning/Hessians into learning rate selection rules (Duchi et al., 2011; Dauphin et al., 2015; Tan et al., 2016). Among numerous proposals, a simple yet widely employed approach is to repeatedly halve the learning rate after performing a *pre-determined* number of iterations (see, for example, Bottou et al., 2018).

In this paper, we introduce a new variant of SGD that we term *SplitSGD* with a novel learning rate selection rule. At a high level, our new method is motivated by the following fact: an optimal learning rate should be adaptive to the *informativeness* of the noisy gradient $g(\theta_t, Z_{t+1})$. Roughly speaking, the informativeness is higher if the true gradient $\nabla F(\theta_t)$ is relatively large compared with the noise

$\nabla F(\theta_t) - g(\theta_t, Z_{t+1})$ and vice versa. On the one hand, if the learning rate is too small with respect to the informativeness of the noisy gradient, SGD makes rather slow progress. On the other hand, the iterates would bounce around a region of an optimum of the objective if the learning rate is too large with respect to the informativeness. The latter case corresponds to a stationary phase in stochastic optimization (Murata, 1998; Chee and Toulis, 2018), which necessitates the *reduction* of the learning rate for better convergence. Specifically, let $\pi_\eta$ be the stationary distribution for $\theta$ when the learning rate is constant and set to $\eta$. From (2) one has that $\mathbb{E}_{\theta \sim \pi_\eta}[g(\theta, Z)] = 0$, and consequently that

$$\mathbb{E}[\langle g(\theta^{(1)}, Z^{(1)}), g(\theta^{(2)}, Z^{(2)})\rangle] = 0 \qquad \text{for} \quad \theta^{(1)}, \theta^{(2)} \overset{i.i.d.}{\sim} \pi_\eta, \quad Z^{(1)}, Z^{(2)} \overset{i.i.d.}{\sim} Z \qquad (3)$$

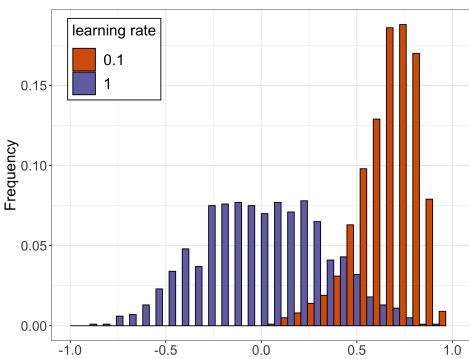

Figure 1: Normalized dot product of averaged noisy gradients over 100 iterations. Stationarity depends on the learning rate: $\eta = 1$ corresponds to stationarity (purple), while $\eta = 0.1$ corresponds to non stationarity (orange). Details in Section 2.

SplitSGD differs from other stochastic optimization procedures in its *robust* stationarity phase detection, which we refer to as the *Splitting Diagnostic*. In short, this diagnostic runs two SGD threads initialized at the same iterate using *independent* data points (refers to $Z_{t+1}$ in (2)), and then performs hypothesis testing to determine whether the learning rate leads to a stationary phase or not. The effectiveness of the Splitting Diagnostic is illustrated in Figure 1, which reveals different patterns of dependence between the two SGD threads with difference learning rates. Loosely speaking, in the stationary phase (in purple), the two SGD threads behave as if they are independent due to a large learning rate, and SplitSGD subsequently decreases the learning rate by some factor. In contrast, strong positive dependence is exhibited in the non stationary phase (in orange) and, thus, the learning rate remains the same after the diagnostic. In essence, the robustness of the Splitting Diagnostic is attributed to its adaptivity to the *local geometry* of the objective, thereby making SplitSGD a *tuning-insensitive* method for stochastic optimization. Its strength is confirmed by our experimental results in both convex and non-convex settings. In the latter, SplitSGD showed robustness with respect to the choice of the initial learning rate, and remarkable success in improving the test accuracy and avoiding overfitting compared to classic optimization procedures.

## 1.1 RELATED WORK

There is a long history of detecting stationarity or non-stationarity in stochastic optimization to improve convergence rates (Yin, 1989; Pflug, 1990; Delyon and Juditsky, 1993; Murata, 1998; Pesme et al., 2020). Perhaps the most relevant work in this vein to the present paper is Chee and Toulis (2018), which builds on top of Pflug (1990) for general convex functions. Specifically, this work uses the running sum of the inner products of successive stochastic gradients for stationarity detection. However, this approach does not take into account the strong correlation between consecutive gradients and, moreover, is not sensitive to the local curvature of the current iterates due to unwanted influence from prior gradients. In contrast, the splitting strategy, which is akin to HiGrad (Su and Zhu, 2018), allows our SplitSGD to concentrate on the current gradients and leverage the regained independence of gradients to test stationarity. Lately, Yaida (2019) and Lang et al. (2019) derive a stationarity detection rule that is based on gradients of a mini-batch to tune the learning rate in SGD with momentum.

From a different angle, another related line of work is concerned with the relationship between the informativeness of gradients and the mini-batch size (Keskar et al., 2016; Yin et al., 2017; Li et al., 2017; Smith et al., 2017). Among others, it has been recognized that the optimal mini-batch size should be adaptive to the local geometry of the objective function and the noise level of the gradients, delivering a growing line of work that leverage the mini-batch gradient variance for learning rate selection (Byrd et al., 2012; Balles et al., 2016; Balles and Hennig, 2017; De et al., 2017; Zhang and Mitliagkas, 2017; McCandlish et al., 2018).

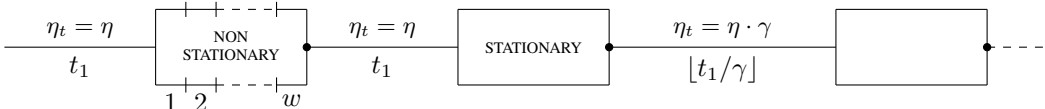

Figure 2: The architecture of SplitSGD. The initial learning rate is $\eta$ and the length of the first single thread is $t_1$. If the diagnostic does not detect stationarity, the length and learning rate of the next thread remain unchanged. If stationarity is observed, we decrease the learning rate by a factor $\gamma$ and proportionally increase the length.

## 2 THE SPLITSGD ALGORITHM

In this section, we first develop the Splitting Diagnostic for stationarity detection, followed by the introduction of the SplitSGD algorithm in detail.

### 2.1 DIAGNOSTIC VIA SPLITTING

Intuitively, the stationarity phase occurs when two independent threads with the same starting point are *no longer* moving along the same direction. This intuition is the motivation for our Splitting Diagnostic, which is presented in Algorithm 1 and described in what follows. We call $\theta_0$ the initial value, even though later it will often have a different subscript based on the number of iterations already computed before starting the diagnostic. From the starting point, we run two SGD threads, each consisting of $w$ windows of length $l$. For each thread $k = 1, 2$, we define $g_t^{(k)} = g(\theta_t^{(k)}, Z_{t+1}^{(k)})$ and the iterates are

$$\theta_{t+1}^{(k)} = \theta_t^{(k)} - \eta \cdot g_t^{(k)}, \tag{4}$$

where $t \in \{0, ..., wl - 1\}$. On every thread we compute the average noisy gradient in each window, indexed by $i = 1, ..., w$, which is

$$\bar{g}_i^{(k)} := \frac{1}{l} \sum_{j=1}^{l} g_{(i-1) \cdot l + j}^{(k)} = \frac{\theta_{(i-1) \cdot l + 1}^{(k)} - \theta_{i \cdot l + 1}^{(k)}}{l \cdot \eta}. \tag{5}$$

The length $l$ of each window has the same function as the mini-batch parameter in mini-batch SGD (Li et al., 2014), in the sense that a larger value of $l$ aims to capture more of the true signal by averaging out the errors. At the end of the diagnostic, we have stored two vectors, each containing the average noisy gradients in the windows in each thread.

**Definition 2.1.** *For $i = 1, ..., w$, we define the gradient coherence with respect to the starting point of the Splitting Diagnostic $\theta_0$, the learning rate $\eta$, and the length of each window $l$, as*

$$Q_i(\theta_0, \eta, l) = \langle \bar{g}_i^{(1)}, \bar{g}_i^{(2)} \rangle. \tag{6}$$

*We will drop the dependence from the parameters and refer to it simply as $Q_i$.*

The gradient coherence expresses the relative position of the average noisy gradients, and its sign indicates whether the SGD updates have reached stationarity. In fact, if in the two threads the noisy gradients are pointing on average in the same direction, it means that the signal is stronger than the noise, and the dynamic is still in its transient phase. On the contrary, as (3) suggests, when the gradient coherence is on average very close to zero, and it also assumes negative values thanks to its stochasticity, this indicates that the noise component in the gradient is now dominant, and stationarity has been reached. Of course these values, no matter how large $l$ is, are subject to some randomness. Our diagnostic then considers the signs of $Q_1, ..., Q_w$ and returns a result based on the proportion of negative $Q_i$. One output is a boolean value $T_D$, defined as follows:

$$T_D = \begin{cases} S & \text{if} \quad \sum_{i=1}^{w} (1 - \text{sign}(Q_i))/2 \geq q \cdot w \\ N & \text{if} \quad \sum_{i=1}^{w} (1 - \text{sign}(Q_i))/2 < q \cdot w. \end{cases} \tag{7}$$

where $T_D = S$ indicates that stationarity has been detected, and $T_D = N$ means non-stationarity. The parameter $q \in [0, 1]$ controls the tightness of this guarantee, being the smallest proportion of negative $Q_i$ required to declare stationarity. In addition to $T_D$, we also return the average last iterate of the two threads as a starting point for following iterations. We call it $\theta_D := (\theta_{w \cdot l}^{(1)} + \theta_{w \cdot l}^{(2)})/2$.

## 2.2 THE ALGORITHM

---

**Algorithm 1** SplitSGD

---

    **SplitSGD**$(\eta, w, l, q, B, t_1, \theta_0, \gamma)$

1: $\eta_1 = \eta$
2: $\theta_1^{in} = \theta_0$
3: **for** $b = 1, ..., B$ **do**
4:     Run SGD with constant step size $\eta_b$ for
        $t_b$ steps, starting from $\theta_b^{in}$
5:     Let the last update be $\theta_b^{last}$
6:     $D_b = $ **Diagnostic**$(\eta_b, w, l, q, \theta_b^{last})$
7:     $\theta_{b+1}^{in} = \theta_{D_b}$
8:     **if** $T_{D_b} = S$ **then**
9:         $\eta_{b+1} = \gamma \cdot \eta_b$ and $t_{b+1} = \lfloor t_b / \gamma \rfloor$
10:    **else**
11:        $\eta_{b+1} = \eta_b$ and $t_{b+1} = t_b$
12:    **end if**
13: **end for**

    **Diagnostic**$(\eta, w, l, q, \theta^{in})$
14: $\theta_0^{(1)} = \theta_0^{(2)} = \theta^{in}$
15: **for** $i = 1, ..., w$ **do**
16:     **for** $k = 1, 2$ **do**
17:         **for** $j = 0, ..., l - 1$ **do**
18:             $ind = (i - 1) \cdot l + j$
19:             $\theta_{ind+1}^{(k)} = \theta_{ind}^{(k)} - \eta \cdot g_{ind}^{(k)}$
20:         **end for**
21:         $\bar{g}_i^{(k)} = (\theta_{(i-1) \cdot l + 1}^{(k)} - \theta_{i \cdot l}^{(k)}) / l \cdot \eta$.
22:     **end for**
23:     $Q_i = \langle \bar{g}_i^{(1)}, \bar{g}_i^{(2)} \rangle$
24: **end for**
25: **if** $\sum_{i=1}^{w} (1 - \text{sign}(Q_i))/2 \geq q \cdot w$ **then**
26:     **return** $\left\{ \theta_D = (\theta_{w \cdot l}^{(1)} + \theta_{w \cdot l}^{(2)})/2, T_D = S \right\}$
27: **else**
28:     **return** $\left\{ \theta_D = (\theta_{w \cdot l}^{(1)} + \theta_{w \cdot l}^{(2)})/2, T_D = N \right\}$
29: **end if**

---

The Splitting Diagnostic can be employed in a more sophisticated SGD procedure, which we call SplitSGD. We start by running the standard SGD with constant learning rate $\eta$ for $t_1$ iterations. Then, starting from $\theta_{t_1}$, we use the Splitting Diagnostic to verify if stationarity has been reached. If stationarity is not detected, the next single thread has the same length $t_1$ and learning rate $\eta$ as the previous one. On the contrary, if $T_D = S$, we decrease the learning rate by a factor $\gamma \in (0, 1)$ and increase the length of the thread by $1/\gamma$, as suggested by Bottou et al. (2018) in their SGD$^{1/2}$ procedure. Notice that, if $q = 0$, then the learning rate gets deterministically decreased after each diagnostic. On the other extreme, if we set $q = 1$, then the procedure maintains constant learning rate with high probability. Figure 2 illustrates what happens when the first diagnostic does not detect stationarity, but the second one does. SplitSGD puts together two crucial aspects: it employs the Splitting Diagnostic at deterministic times, but it does not deterministically decreases the learning rate. We will see in Section 4 how both of these features come into play in the comparison with other existing methods. A detailed explanation of SplitSGD is presented in Algorithm 1.

## 3 THEORETICAL GUARANTEES FOR STATIONARITY DETECTION

This section develops theoretical guarantees for the validity of our learning rate selection. Specifically, in the case of a relatively small learning rate, we can imagine that, if the number of iterations is fixed, the SGD updates are not too far from the starting point, so the stationary phase has not been reached yet. On the other hand, however, when $t \to \infty$ and the learning rate is fixed, we would like the diagnostic to tell us that we have reached stationarity, since we know that in this case the updates will oscillate around $\theta^*$. Our first assumption concerns the convexity of the function $F(\theta)$. It will not be used in Theorem 3.1, in which we focus our attention on a neighborhood of $\theta_0$.

**Assumption 3.1.** *The function $F$ is strongly convex, with convexity constant $\mu > 0$. For all $\theta_1, \theta_2$,*

$$F(\theta_1) \geq F(\theta_2) + \langle \nabla F(\theta_2), \theta_1 - \theta_2 \rangle + \frac{\mu}{2} \|\theta_1 - \theta_2\|^2$$

*and also $\|\nabla F(\theta_1) - \nabla F(\theta_2)\| \geq \mu \cdot \|\theta_1 - \theta_2\|$.*

**Assumption 3.2.** *The function $F$ is smooth, with smoothness parameter $L > 0$. For all $\theta_1, \theta_2$,*

$$\|\nabla F(\theta_1) - \nabla F(\theta_2)\| \leq L \cdot \|\theta_1 - \theta_2\|.$$

We said before that the noisy gradient is an unbiased estimate of the true gradient. The next assumption that we make is on the distribution of the errors.

**Assumption 3.3.** *We define the error in the evaluation of the gradient in $\theta_{t-1}$ as*

$$\epsilon_t := \epsilon(\theta_{t-1}, Z_t) = g(\theta_{t-1}, Z_t) - \nabla F(\theta_{t-1}) \tag{8}$$

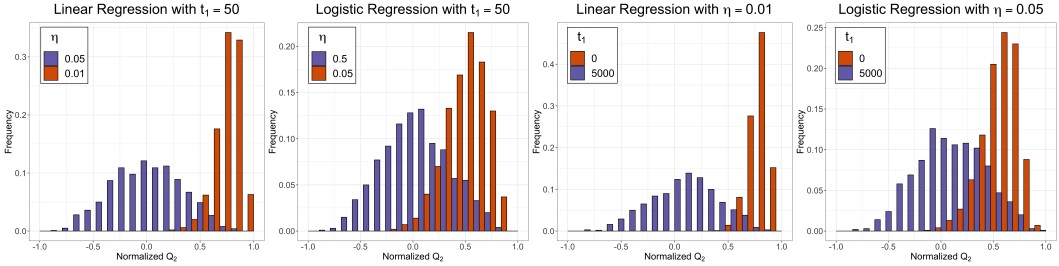

Figure 3: Histogram of the gradient coherence $Q_i$ (for the second pair of windows, normalized) of the Splitting Diagnostic for linear and logistic regression. The two left panels show the behavior in Theorem 3.1, the two right panels the one in Theorem 3.2. In orange we see non stationarity, while in purple a distribution that will return stationarity for an appropriate choice of $w$ and $q$.

and the filtration $\mathcal{F}_t = \sigma(Z_1, ..., Z_t)$. Then $\epsilon_t \in \mathcal{F}_t$ and $\{\epsilon_t\}_{t=1}^\infty$ is a martingale difference sequence with respect to $\{\mathcal{F}_t\}_{t=1}^\infty$, which means that $\mathbb{E}[\epsilon_t|\mathcal{F}_{t-1}] = 0$. The covariance of the errors satisfies

$$\sigma_{\min} \cdot I \preceq \mathbb{E}\left[\epsilon_t \epsilon_t^T \mid \mathcal{F}_{t-1}\right] \preceq \sigma_{\max} \cdot I, \tag{9}$$

where $0 < \sigma_{\min} \leq \sigma_{\max} < \infty$ for any $\theta$.

Our last assumption is on the noisy functions $f(\theta, Z)$ and on an upper bound on the moments of their gradient. We do not specify $m$ here since different values are used in the next two theorems, but the range for this parameter is $m \in \{2, 4\}$.

**Assumption 3.4.** *Each function $f(\theta, Z)$ is convex, and there exists a constant $G$ such that $\mathbb{E}\left[\|g(\theta_t, Z_{t+1})\|^m \mid \mathcal{F}_t\right] \leq G^m$ for any $\theta_t$.*

We first show that there exists a learning rate sufficiently small such that the standard deviation of any gradient coherence $Q_i$ is arbitrarily small compared to its expectation, and the expectation is positive because $\theta_{t_1+l}$ is not very far from $\theta_0$. This implies that the probability of any gradient coherence to be negative, $\mathbb{P}(Q_i < 0)$, is extremely small, which means that the Splitting Diagnostic will return $T_D = N$ with high probability.

**Theorem 3.1.** *If Assumptions 3.2, 3.3 and 3.4 with $m = 4$ hold, $\|\nabla F(\theta_0)\| > 0$ and we run $t_1$ iterations before the Splitting Diagnostic, then for any $i \in \{1, ..., w\}$ we can set $\eta$ small enough to guarantee that*

$$\mathrm{sd}(Q_i) \leq C_1(\eta, l) \cdot \mathbb{E}[Q_i],$$

*where $C_1(\eta, l) = O(1/\sqrt{l}) + O(\sqrt{\eta(t_1 + l)})$. Proof in Appendix B.*

In the two left panels of Figure 3 we provide a visual interpretation of this result. When the starting point of the SGD thread is sufficiently far from the minimizer $\theta^*$ and $\eta$ is sufficiently small, then all the mass of the distribution of $Q_i$ is concentrated on positive values, meaning that the Splitting Diagnostic will not detect stationarity with high probability. In particular we can use Chebyshev inequality to get a bound for $\mathbb{P}(Q_i < 0)$ of the following form:

$$\mathbb{P}(Q_i < 0) \leq \mathbb{P}(|Q_i - \mathbb{E}[Q_i]| > \mathbb{E}[Q_i]) \leq \mathrm{sd}(Q_i)^2/\mathbb{E}[Q_i]^2 \leq C_1(\eta, l)^2$$

Note that to prove Theorem 3.1 we do not need to use the strong convexity Assumption 3.1 since, when $\eta(t_1 + l)$ is small, $\theta_{t_1+l}$ is not very far from $\theta_0$. In the next Theorem we show that, if we let the SGD thread before the diagnostic run for long enough and the learning rate is not too big, then the splitting diagnostic output is $T_D = S$ probability that can be made arbitrarily high. This is consistent with the fact that, as $t_1 \to \infty$, the iterates will start oscillating in a neighborhood of $\theta^*$.

**Theorem 3.2.** *If Assumptions 3.1, 3.2, 3.3 and 3.4 with $m = 2$ hold, then for any $\eta \leq \frac{\mu}{L^2}$, $l \in \mathbb{N}$ and $i \in \{1, ..., w\}$, as $t_1 \to \infty$ we have*

$$|\mathbb{E}[Q_i]| \leq C_2(\eta) \cdot \mathrm{sd}(Q_i),$$

*where $C_2(\eta) = C_2 \cdot \eta + o(\eta)$. Proof in Appendix C.*

The result of this theorem is confirmed by what we see in the right panels of Figure 3. There, most of the mass of $Q_i$ is on positive values if $t_1 = 0$, since the learning rate is sufficiently small and the

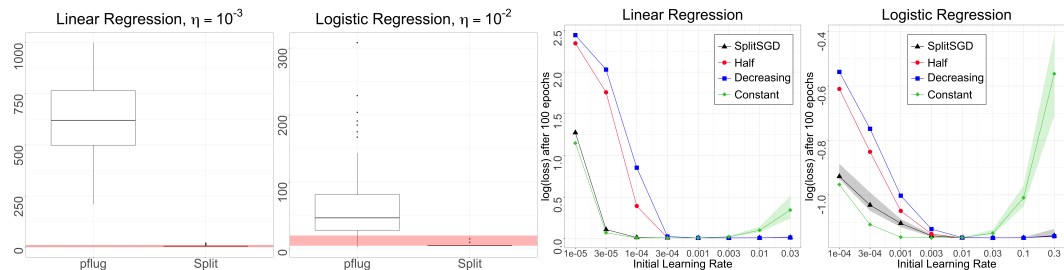

Figure 4: (left) comparison between Splitting and *pflug* Diagnostics on linear and logistic regression. The red bands are the epochs where stationarity should be detected. (right) comparison of the log(loss) achieved after 100 epochs between SplitSGD, SGD$^{1/2}$ (Half) and SGD with constant or decreasing learning rate on linear and logistic regression. More details are in Section 4.1.

starting point is not too close to the minimizer. But when we let the first thread run for longer, we see that the distribution of $Q_i$ is now centered around zero, with an expectation that is much smaller than its standard deviation. An appropriate choice of $w$ and $q$ makes the probability that $T_D = S$ arbitrarily big. In the proof of Theorem 3.2, we make use of a result that is contained in Moulines and Bach (2011) and then subsequently improved in Needell et al. (2014), representing the dynamic of SGD with constant learning rate.

**Lemma 3.3.** *If Assumptions 3.1, 3.2, 3.3 and 3.4 with $m = 2$ hold, and $\eta \leq \frac{\mu}{L^2}$, then for any $t \geq 0$*

$$\mathbb{E}\left[\|\theta_t - \theta^*\|^2\right] \leq \left(1 - 2\eta(\mu - L^2\eta)\right)^t \cdot \mathbb{E}\left[\|\theta_0 - \theta^*\|^2\right] + \frac{G^2\eta}{\mu - L^2\eta}.$$

The simulations in Figure 3 show us that, once stationarity is reached, the distribution of the gradient coherence is fairly symmetric and centered around zero, so its sign will be approximately a coin flip. In this situation, if $l$ is large enough, the count of negative gradient coherences is approximately distributed as a Binomial with $w$ number of trials, and $0.5$ probability of success. Then we can set $q$ to control the probability of making a type I error – rejecting stationarity after it has been reached – by making $\frac{1}{2^w}\sum_{i=0}^{q \cdot w - 1}\binom{w}{i}$ sufficiently small. Notice that a very small value for $q$ makes the type I error rate decrease but makes it easier to think that stationarity has been reached too early. In the Appendix E.1 we provide a simple visual interpretation to understand why this trade-off gets weaker as $w$ becomes larger. Finally, we provide a result on the convergence of SplitSGD. We leave for future work to prove the convergence rate of SplitSGD, which appears to be a very challenging problem.

**Proposition 3.4.** *If Assumptions 3.1, 3.2, 3.3 and 3.4 with $m = 2$ hold, and $\eta \leq \frac{\mu}{L^2}$, then SplitSGD is guaranteed to converge with probability tending to 1 as the number of diagnostics $B \to \infty$. Proof in Appendix D.*

# 4 EXPERIMENTS

## 4.1 CONVEX OBJECTIVE

The setting is described in details in Appendix E.1. We use a feature matrix $X \in \mathbb{R}^{n \times d}$ with standard normal entries and $n = 1000$, $d = 20$ and $\theta_j^* = 5 \cdot e^{-j/2}$ for $j = 1, ..., 20$. The key parameters are $t_1 = 4, w = 20, l = 50$ and $q = 0.4$. A sensitivity analysis is in Section 4.3.

**Comparison between splitting and `pflug` diagnostic.** In the left panels of Figure 4 we compare the Splitting Diagnostic with the `pflug` Diagnostic introduced in Chee and Toulis (2018). The boxplots are obtained running both diagnostic procedures from a starting point $\theta_0 = \theta_s + \epsilon'$, where $\epsilon' \sim N(0, 0.01I_d)$ is multivariate Gaussian and $\theta_s$ has the same entries of $\theta^*$ but in reversed order, so $\theta_{s,j} = 5 \cdot e^{-(d-j)/2}$ for $j = 1, ..., 20$. Each experiment is repeated 100 times. For the Splitting Diagnostic, we run SplitSGD and declare that stationarity has been detected at the first time that a diagnostic gives result $T_D = S$, and output the number of epochs up to that time. For the `pflug` diagnostic, we stop when the running sum of dot products used in the procedure becomes negative at the end of an epoch. The maximum number of epochs is 1000, and the red horizontal bands represent

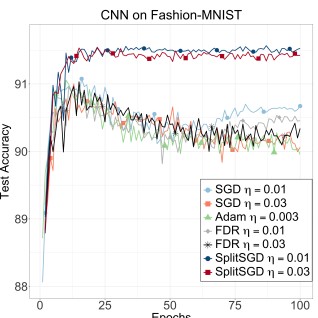 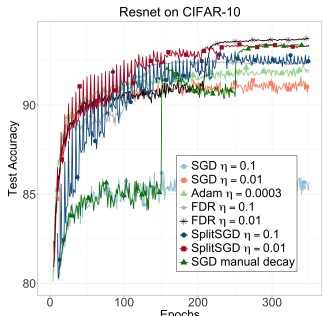 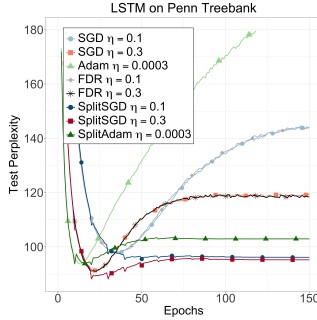

Figure 5: Performance of SGD, Adam, FDR and SplitSGD in training different neural networks. SplitSGD proved to be beneficial in (i) better robustness to the choice of initial learning rates, (ii) achieving higher test accuracy when possible, and (iii) reducing the effect of overfitting. Details of each plot are in Section 4.2.

the approximate values for when we can assume that stationarity has been reached, based on when the loss function of SGD with constant learning rate stops decreasing. We can see that the result of the Splitting Diagnostic is close to the truth, while the pflug Diagnostic incurs the risk of waiting for too long, when the initial dot products of consecutive noisy gradients are positive and large compared to the negative increments after stationarity is reached. The Splitting Diagnostic does not have this problem, as a checkpoint is set every fixed number of iterations. The previous computations are then discarded, and only the new learning rate and starting point are stored. In Appendix E.2 we show more configurations of learning rates and starting points.

**Comparison between SplitSGD and other optimization procedures.** Here we set the decay rate to the standard value $\gamma = 0.5$, and compare SplitSGD with SGD with constant learning rate $\eta$, SGD with decreasing learning rate $\eta_t \propto 1/\sqrt{t}$ (where the initial learning rate is set to $20\eta$), and SGD$^{1/2}$ (Bottou et al., 2018), where the learning rate is halved deterministically and the length of the next thread is double that of the previous one. For SGD$^{1/2}$ we set the length of the initial thread to be $t_1$, the same as for SplitSGD. In the right panels of Figure 4 we report the log of the loss that we achieve after 100 epochs for different choices of the initial learning rate. It is clear that keeping the learning rate constant is optimal when its initial value is small, but becomes problematic for large initial values. On the contrary, deterministic decay can work well for larger initial learning rates but performs poorly when the initial value is small. Here, SplitSGD shows its robustness with respect to the initial choice of the learning rate, performing well on a wide range of initial learning rates.

## 4.2 DEEP NEURAL NETWORKS

To train deep neural networks, instead of using the simple SGD with a constant learning rate inside the SplitSGD procedure, we adopt SGD with momentum (Qian, 1999), where the momentum parameter is set to 0.9. SGD with momentum is a popular choice in training deep neural networks (Sutskever et al., 2013), and when the learning rate is constant, it still exhibits both transient and stationary phase. We introduce three more differences with respect to the convex setting: (i) the gradient coherences are defined for each layer of the network separately, then counted together to globally decay the learning rate for the whole network, (ii) the length of the single thread is not increased if stationarity is detected, and (iii) we consider the default parameters $q = 0.25$ and $w = 4$ for each layer. We expand on these differences in Appendix E.3. As before, the length of the Diagnostic is set to be one epoch, and $t_1 = 4$. We compare SplitSGD with SGD with momentum, Adam (Kingma and Ba, 2014) and FDR (Yaida, 2019) with different learning rates, and report the ones that show the best results. Notice that, although $\eta = 3e-4$ is the popular default value for Adam, this method is still sensitive to the choice of the learning rate, so the best performance can be achieved with other values. For FDR, we tested each setting with the parameter t_adaptive $\in \{100, 1000\}$, which gave similar results. It has also been proved that SGD generalizes better than Adam (Keskar and Socher, 2017; Luo et al., 2019). We show that in many situations SplitSGD, using the same default parameters, can outperform both. In Figure 5 we report the average results of 5 runs. In Figure 10 in the appendix we consider the same plot but also add 90% confidence bands, omitted here for better readability.

**Convolutional neural networks (CNNs).** We consider a CNN with two convolutional layers and a final linear layer trained on the Fashion-MNIST dataset (Xiao et al., 2017). We set $\eta \in \{1e-2, 3e-2, 1e-1\}$ for SGD and SplitSGD, $\eta \in \{1e-2, 1e-1\}$ for FDR and $\eta \in \{3e-4, 1e-3, 3e-3, 1e-2\}$ for Adam. In the first panel of Figure 5 we see the interesting fact that SGD, FDR and Adam all show clear signs of overfitting, after reaching their peak in the first 20 epochs. SplitSGD, on the contrary, does not incur in this problem, but for a combined effect of the averaging and learning rate decay is able to reach a better overall performance without overfitting. We also notice that SplitSGD is very robust with respect to the choice of the initial learning rate, and that its peak performance is better than the one of any of the competitors.

**Residual neural networks (ResNets).** For ResNets, we consider a 18-layer ResNet[1] and evaluate it on the CIFAR-10 dataset (Krizhevsky et al., 2009). We use the initial learning rates $\eta \in \{1e-3, 1e-2, 1e-1\}$ for SGD and SplitSGD, $\eta \in \{1e-2, 1e-1\}$ for FDR and $\eta \in \{3e-5, 3e-4, 3e-3\}$ for Adam, and also consider the SGD procedure with manual decay that consists in setting $\eta = 1e-1$ and then decreasing it by a factor 10 at epoch 150 and 250. In the second panel of Figure 5 we clearly see a classic behavior for SplitSGD. The averaging after the diagnostics makes the test accuracy peak, but the improvement is only momentary as the learning rate is not decreased. When the decay happens, the peak is maintained and the fluctuations get smaller. We can see that SplitSGD, with both initial learning rate $\eta = 1e-2$ and $\eta = 1e-1$ is better than both SGD and Adam and that one setting achieves the same final test accuracy of the manually tuned method in less epochs. The FDR method is showing excellent performance when $\eta = 0.01$ and a worse result when $\eta = 0.1$. In Appendix E.4 we see a similar plot obtained with the neural network VGG19.

**Recurrent neural networks (RNNs).** For RNNs, we evaluate a two-layer LSTM (Hochreiter and Schmidhuber, 1997) model on the Penn Treebank (Marcus et al., 1993) language modelling task. We use $\eta \in \{0.1, 0.3, 1.0\}$ for both SGD and SplitSGD, $\eta \in \{0.1, 0.3\}$ for FDR, $\eta \in \{1e-4, 3e-4, 1e-3\}$ for Adam and also introduce SplitAdam, a method similar to SplitSGD, but with Adam in place of SGD with momentum. As shown in the third panel of Figure 5, we can see that SplitSGD outperforms SGD and SplitAdam outperforms Adam with regard to both the best performance and the last performance. FDR is not showing any improvement compared to standard SGD, meaning that in this framework it is unable to detect stationarity and decay the learning rate accordingly. Similar to what already observed with the CNN, we need to note that our proposed splitting strategy has the advantage of reducing the effect of overfitting, which is very severe for SGD, Adam and FDR while very small for SplitAdam and SplitSGD. We postpone the theoretical understanding for this phenomena as our future work.

For the deep neural networks considered here, SplitSGD shows better results compared to SGD and Adam, and exhibits strong robustness to the choice of initial learning rates, which further verifies the effectiveness of SplitSGD in deep neural networks. The Splitting Diagnostic is proved to be beneficial in all these different settings, reducing the learning rate to enhance the test performance and reduce overfitting of the networks. FDR shows a good result when used on ResNet with a specific learning rate, but in the other setting is not improving over SGD, suggesting that its diagnostic does not work on a variety of different scenarios.

### 4.3 SENSITIVITY ANALYSIS FOR SPLITSGD

In this section, we analyse the impact of the hyper-parameters in the SplitSGD procedure. We focus on $q$ and $w$, while $l$ changes so that the computational budget of each diagnostic is fixed at one epoch. In the left panels of Figure 6 we analyse the sensitivity of SplitSGD to these two parameters in the convex setting, for both linear and logistic regression, and consider $w \in \{10, 20, 40\}$ and $q \in \{0.35, 0.40, 0.45\}$. The data are generated in the same way as those used in Section 4.1. On the y-axis we report the log(loss) after training for 100 epochs, while on the x-axis we consider the different $(w, q)$ configurations. The results are as expected; when the initial learning rate is larger, the impact of these parameters is very modest. When the initial learning rate is small, having a quicker decay (i.e. setting $q$ smaller) worsen the performance.

In the right panels of Figure 6 we see the same analysis applied to the FeedForward Neural Network (FNN) described in Appendix E.4 and the CNN used before, both trained on Fashion-MNIST. Here

---

[1]More details can be found in `https://pytorch.org/docs/stable/torchvision/models.html`.

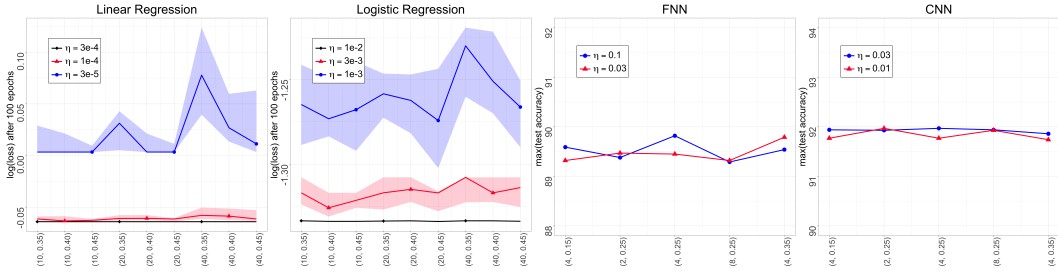

Figure 6: Sensitivity analysis for SplitSGD with respect to the parameters $w$ and $q$, appearing as the labels of the x-axis in the form $(w, q)$. In the convex setting (left) we consider the log loss achieved after 100 epochs, while for deep neural networks (right) we report the maximum of the test accuracy. Details in Section 4.3.

we report the maximum test accuracy achieved when training for 100 epochs, and on the x-axis we have various configurations for $q \in \{0.15, 0, 25, 0.35\}$ and $w \in \{2, 4, 8\}$. The results are very encouraging, showing that SplitSGD is robust with respect to the choice of these parameters also in non-convex settings.

## 5 CONCLUSION AND FUTURE WORK

We have developed an efficient optimization method called SplitSGD, by splitting the SGD thread for stationarity detection. Extensive simulation studies show that this method is robust to the choice of the initial learning rate in a variety of optimization tasks, compared to classic non-adaptive methods. Moreover, SplitSGD on certain deep neural network architectures outperforms classic SGD, Adam and FDR in terms of the test accuracy, and can sometime limit greatly the impact of overfitting. As the critical element underlying SplitSGD, the Splitting Diagnostic is a simple yet effective strategy that can possibly be incorporated into many optimization methods beyond SGD, as we already showed training SplitAdam on LSTM. One possible limitation of this method is the introduction of a new relevant parameter $q$, that regulates the rate at which the learning rate is adaptively decreased. Our simulations suggest the use of two different values depending on the context. A slower decrease, $q = 0.4$, in convex optimization, and a more aggressive one, $q = 0.25$, for deep learning. In the future, we look forward to seeing research investigations toward boosting the convergence of SplitSGD by allowing for different learning rate selection strategies across different layers of the neural networks.

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
