# OpenReview forum: "Robust Learning Rate Selection for Stochastic Optimization via Splitting Diagnostic"
_ICLR.cc/2021/Conference — Reject_

### Official Review · AnonReviewer4 · 2020-10-27
**Robust Learning Rate Selection for Stochastic Optimization via Splitting Diagnostic**

**Rating:** 3
**Confidence:** 4

**Review:**

Review: This paper proposes SplitSGD, a novel heuristic for adapting the learning rate. This novel learning rate schedule is based
on the identification of stationary phases, where for stationary phases the authors refer to the training stage where the noise
in the gradient estimate becomes dominant and therefore the iterates start jumping around a stationary point.
The authors develop an heuristic to detect such phases. When a stationary phase is detected, the learning rate is
decreased of a factor $\gamma$. Under some assumptions, the authors provide a theoretical analysis for the
stationary phase detection mechanism which also underlines the relation with the learning rate value. SplitSGD is then benchmarked and compared against some of the standard learning rate decaying heuristics generally adopted
in combination with SGD or other first order stochastic methods.
___________________________________________________________________________________________________________________

+ Overall the paper is well-written and clear.

___________________________________________________________________________________________________________________

Concerns:

1. The learning rate, also called step-size, can be interpreted as a brutal approximation of the local curvature with a scaled identity matrix. All the work on second-order methods attempts to refine this brutal approximation with better estimates of the local curvature. Despite being the learning rate and how to adjust it the central topic of this paper, nothing is mentioned regarding the
fundamental relation of the learning rate and the local curvature.


2. There are some wrong statements here and there in the paper, i.e.
"""
Specifically, in the case of a relatively small learning rate, we can imagine that, if the number of iterations is fixed, the SGD updates are not too far from the starting point, so the stationary phase has not been reached yet.
 """


3. The authors claim to propose an optimization method while what they are proposing is an heuristic to decrease the learning rate in an adaptive fashion.


4. The costs of the stationary phase check are not discussed but the authors just briefly mention that SplitSGD comes with no significant extra costs. This does not seem to be the case though.


5. The literature on increasing the batch size toward the final part of training is not discussed at all, as well as other state-of-the-art heuristics to deal with the noisy gradient estimate.


6. The benchmarks are limited as the authors are not considering state-of-the-art learning rate  decaying heuristics such as the cosine decaying schedule. In addition, they could show more solid empirical results by letting an hyperparameter optimizer such as BOHB choose for the best learning schedule. Otherwise it is hard to say that the superior performance of SplitSGD is not a consequence of a wrong hyperparameter tuning of the competitors.


7. I think the deep learning community at this stage needs less work on learning rate heuristics and more attentions on the theoretical analysis of the geometry of the landscape, novel optimization methods with theoretical guarantees and solid work on generalization properties, as there has been enough works on such heuristics which often ends up 'adding extra noise' and therefore constitute an obstacle in the process of bringing clarity and understanding in the field.

---

> ### Author Response · Authors · 2020-11-12
> **Response to Reviewer #4**
>
> Dear Reviewer #4, thanks for undertaking the review of our paper and providing comments!
>
> Before giving the item by item responses below, we would like to point out that both reviewer #1 and reviewer #3 highlighted the **novelty** of our work, its **theoretical validity** and the **strength of the empirical evaluations** that we performed. In particular, the reviewers and we the authors all believe that simple and effective learning rate schedules are crucial to the success of deep learning research and, therefore, more efforts are needed in this research direction, as opposed to point 7 below.
>
> Moreover, we would like to add that, even outside of the framework of learning rate adaptation, our diagnostic for detecting stationarity could be an interesting technique to be applied for a **variety of settings** where deep learning is used. We believe that our work opens a venue for future research making use of statistical ideas for improving effectiveness and robustness of deep learning models. We hope that our work can be judged with these appealing features being recognized.
>
> We would like now to respond to the other points that have been raised:
>
> 1. This is a very good point and we will add to the introduction the connection between the learning rate and the Hessian matrix.
> 2. The highlighted statement needs further clarification. It is in fact necessary to add that we are assuming that the starting point $\theta_0$ is sufficiently far away from the minimizer $\theta^*$, where “sufficiently” also depends on the size of the learning rate. In the strongly convex setting, in fact, this is enough to guarantee that Theorem 3.1 holds and the two threads of the splitting diagnostic will have positive inner product with high probability.
> 3. We believe that selection of learning rate is *critical* in optimization. An effective choice of this parameter can make itself an optimization method. For example, the celebrated *Barzilai-Borwein* actually boils down to selecting a step size for the steepest descent method.
> 4. The splitting diagnostic uses a **fixed number of updates** $2\cdot l\cdot w$ (which in the applications is set to be one epoch). If we were not to run the diagnostic, those updates could be used in a single thread of length $2\cdot l\cdot w$, instead of two threads each of length $l\cdot w$. This difference is clear in the right panels of figure 4, where for very small values of the learning rate SplitSGD is performing slightly worse than SGD with constant learning rate. This is because if the diagnostic never detects stationarity, then the two parallel threads get close to the minimizer slower than a single thread of double length. **The only extra computational cost**, except for this allocation of the budget in two threads, **is the computation of the gradient coherences** $Q_i$, which is very small compared to the cost of training the model.
> 5. This is an interesting topic but not directly related to our method. We will add some reference in the related work section.
> 6. We compare SplitSGD against *state-of-the-art* adaptive methods such as Adam and FDR by Yaida (2019), together with a standard schedule for manual decay for SGD on ResNet. The cosine decaying schedule is certainly interesting but it is *not adaptive* and we decided not to compare against this method, since it would also require the tuning of its hyperparameters. For the same reason, we decided to use the default value of the hyperparameters for Adam and FDR (but for the latter we tried two different values for t_adaptive). Notice that in all the deep learning experiments we maintained the **same default set of hyperparameters for SplitSGD** too, instead of tuning the best values of q and w for each scenario, and for this reason we argue that the comparison is fair.
> We also do not think that the clear overfitting shown by SGD and Adam on both CNN on Fashion-MNIST and LSTM on Penn Treebank can be attributed to the lack of tuning of Adam’s hyperparameters, so it is hard to blame on that SplitSGD’s superior performance in these settings.

---

> > ### Comment · AnonReviewer4 · 2020-11-18
> > **Thank you for your clarifications - final review**
> >
> > Dear authors,
> >
> > Thank you for taking the time to write an item by item response. Unfortunately, after a careful consideration of your response, I am not increasing the vote for this paper mainly for the following reasons:
> >
> > 1. This is a very good point and we will add to the introduction the connection between the learning rate and the Hessian matrix.
> >
> > This connection is so important and fundamental that adding few lines in the introduction is definitely **not enough**. Further work should be done in this direction.
> >
> > 3. We believe that selection of learning rate is critical in optimization. An effective choice of this parameter can make itself an optimization method.
> >
> > Then please provide a theoretical analysis of the convergence rate for your optimization method, namely SGD+your adaptive step size heuristic and, if possible, compare it at least with vanilla SGD with a decaying step size.
> >
> > 5. This is an interesting topic but not directly related to our method. We will add some reference in the related work section.
> >
> > I strongly believe this line of research to be closely related to your heuristic for the learning rate adaptation and I believe that given the importance of this line of work you should at least briefly discuss about it and compare against it in your benchmarks.
> >
> > 6.  it is hard to blame on that SplitSGD’s superior performance in these settings.
> >
> > In my opinion it is very hard to claim that SplitSGD enjoys superior performance in terms of convergence without any theoretical analysis of its convergence rate and clear theoretical comparison and only based on the proposed benchmarks. At this level it could be a pure speculation based on few empirical observations. In addition, all the considerations on generalization properties and overfitting are in this context pure speculations. Finally, even if Adam makes use of an adaptive strategy for the step size adaptation, it is also true that, as all the first order methods, it is really sensitive to its hyperparameters. Therefore, for a fair and strong comparison and to make more solid claims I think it is fundamental to use a proper tool for tuning the hyperparameters.
> >
> > Last but not least, the assumptions made are quite unrealistic when dealing with deep learning problems and therefore with non-convex landscapes. Future work should definitely focus on extending the theoretical results and relaxing these assumptions, if the goal is that of focusing application-wise on training of NNs. Finally, for all the cases where convexity holds, the authors should compare their method against state-of-the-art second order methods for the reasons mentioned in point 1 of the review.

---

> > > ### Author Response · Authors · 2020-11-19
> > > **Response to Reviewer #4**
> > >
> > > Dear Reviewer #4, thanks for your reply and for further explaining your point. We would like to add a couple of further elaborations. Given space and time limit, we’ve and will try our best to incorporate your very helpful suggestions and comments in the revision, which have already significantly improved the presentation of our paper. We’d like to thank you for all and hope our revised paper can be judged again.
> > >
> > >
> > > 1. First of all, we want to emphasize that the learning rate and the estimation of the curve are **two different** (almost orthogonal) problems and the present paper mostly focuses on the first one. In its perhaps the most general form, consider the update $\theta_{t+1} = \theta_{t} - \eta\cdot G$, where $\eta$ is the learning rate and **G** is an estimate of the gradient (perhaps having or having not incorporated the curvature). An extensive line of work considers how we can incorporate the Hessian info in a computationally cheap way into G. Our paper, however, focuses on an entirely different angle: we have argued in the paper that it is also important to study eta and in particular how one can choose eta in an adaptive way. In this spirit, our work is **orthogonal** to the existing work that focuses on the Hessian. Having said this, we believe it is a promising direction to combine the strength of both the existing Hessian works and our learning rate strategy and we hope to explore more in this direction (thank you for pointing out the importance and potential). For the current paper, it seems to us that it is difficult include this extension due to the page limit.
> > >
> > >
> > > 3. First of all, we would like to thank you for offering suggestions that would definitely improve our work in the future. In general, it is our impression that many highly influential works on the optimization aspects of neural networks published in ICLR or other top ML venues **do not** come with proofs of the convergence guarantees. For example, Kingma and Ba (2015) only provided a regret analysis for their celebrated and widely used method Adam that was later weakened by the popular result of “Reddi et al. (2019) On the convergence of Adam and beyond”, showing that Adam can have non-zero average regret and even not converge to an optimal solution in a convex optimization setting. The FDR method by Yaida (2019) comes with a theoretical analysis of the fluctuation equations but no convergence guarantees, and the convergence diagnostic by Chee and Toulis (2018) is also not guaranteed to converge (since the convergence they reference in the title is the convergence to the stationary phase). As we explained to Reviewer #1 there are several reasons that make this task very challenging, though we have **strong guarantees** for the stationarity detection rule in our paper (Theorems 3.1 and 3.2). Nevertheless, we believe that it is **very important** to develop rigorous and theoretical guarantees for the method, and we will work on this in a follow-up paper. Again, thank you for pointing out the value of this direction for follow-up work.
> > >
> > >
> > > 5. Thanks for bringing up the related works.  On second thought, now we realize that this is **very** related to our work. This is because essentially the choice of the learning rate is to reflect how much noise is contained, while the noise level is certainly impacted by the batch size. We will discuss the connection in more detail in the revision. Thanks again!
> > >
> > >
> > > 6. Thanks for your concern that have motivated us to improve our work. Specifically, we now have improved our empirical evaluation by running each of the simulations in Figure 5 for five times, reporting the average test result and also adding a plot in the appendix that include 90% confidence bands. All of our claims previously made on superior test accuracy, robustness to the initial choice of the learning rate and nearly complete lack of overfitting still stand. We do agree that the lack of theoretical analysis on the absence of overfitting exhibited by SplitSGD makes those specific claims not general, and we did not intend to assert that this is a general property of SplitSGD in all settings. We do think, however, that it is extremely interesting that in two settings (CNN on Fashion-MNIST and LSTM on Penn Treebank) some classical methods show signs of heavy overfitting, while SplitSGD appears to suffer much less from it. What we provide is a possible explanation for this phenomenon that involves a combined effect of the averaging and learning rate decay.

---

### Official Review · AnonReviewer2 · 2020-10-28
**No clear advantage over existing methods.**

**Rating:** 5
**Confidence:** 4

**Review:**

This paper proposes a sign-based test to determine if a stochastic process is in its stationary state or not. Unlike Pflug test, this test uses two independent trajectories to build its test. It divides each trajectory into w parts and averages the gradients insides each part. Then measure the similarity of each average of one trajectory to the corresponding average from the other trajectory via dot product and then counting the negative signs and positive signs. When the negative signs are above a threshold then the process is its stationary station. Then it shrinks the step size of SGD for future iterations.


Comments:
1- Several parts of the paper needs more clarification and explanations:
It is not clear what the statistical intuition behind the splitting test is and why we should expect a more robust test.
The notion of informativeness is mentioned in the introduction but it is not used in the rest of the paper. Besides in its definition, it doesn’t say how they measure the “relatively largeness” of gradient w.r.t. the noise.
Fig 3 just represent eq 3 and gives no extra information. If it means something else such as the inner product for nonstationary is positive with high probability it should be mentioned clearly
 In related work: 1- Le Roux et. al. don’t detect stationarity. Why do you mention it? 2-  This work “Scott  Pesme,  Aymeric  Dieuleveut,  and  Nicolas  Flammarion.   On  convergence-diagnostic
based step sizes for stochastic gradient descent. ICML20” is highly relevant which is not mentioned.
In assumption 4, there is a parameter m which is not clear what it is or there is a limit on it, for example, can m be infinity?. In Thm 3.1. It doesn’t say what sd(Q_i) is.
It is not clear the connection between the result of Thm, 3.1. And the line above it which says with high probability P(Q_i <0) is small. It would be nice to mention how small the \eta should be to guarantee the ineq. in this theorem. Besides we know that in a stationary state E(Q_i) is zero ( assuming after t_i steps we are in the stationary state ). Then what does this result mean?
For both Thm’s 3.1 and 3.2 it would be helpful to add an interpretation for their result.
2- For the empirical results
I think you should compare against  “Lang, H., Zhang, P., and Xiao, L. (2019). Using statistics to automate stochastic optimization” instead of FDR since FDR test has high variance.
You should add a comparison against “Scott  Pesme,  Aymeric  Dieuleveut,  and  Nicolas  Flammarion.   On convergence-diagnostic based step sizes for stochastic gradient descent” which is a distance-based test instead of sign-based.
Fig 4 shows that your test does not work when the step size is too small. However, all your theories hold when the step size is small. SGD ½ is as good as yours.
For your DNN results, are this results average of several runs? If yes, you should add the std bar into the graph, If not why should we rely on the results?

---

> ### Author Response · Authors · 2020-11-14
> **Response to Reviewer #2**
>
> Dear Reviewer #2, thanks for providing several good comments and suggestions for our paper. Below we will reply to them point by point.
>
> **Reply to “statistical intuition behind splitting test”:** The notion of informativeness mentioned in the introduction is not used directly, but what we measure with the splitting diagnostic is the consequence of the SGD iterates bouncing around the minimizer, which is captured by the gradient coherences $Q_i$. Notice that equation (3) uses two independently sampled points $\theta^i$ from their stationary distribution and two independently sampled noise variables $Z^i$. Figure 3, on the other hand, is meant to represent the behavior of $Q_i$ (normalized) as described in theorem 3.1 and 3.2. What it shows is that **for a sufficiently small learning rate most of the probability mass of** $Q_i$ **is on positive values** (left panels) and that, with the same small values of the learning rate, if we let the number of updates to be sufficiently large then the distribution of $Q_i$ is going to be centered around 0 (right panels). We will add a more detailed description of this behavior in the main text.
>
> **Reply to “relevant work not mentioned”:** Thanks for mentioning the work “Scott Pesme, Aymeric Dieuleveut, and Nicolas Flammarion. On convergence-diagnostic based step sizes for stochastic gradient descent”, we will add it to the related work section.
>
> **Reply to “parameter $m$”:** Thanks for pointing out that we did not clarify the range of values for the parameter m. **It is used in the following theorems as either** $m=2$ **or** $m=4$. Clearly requiring $m=4$ is stronger than requiring $m=2$, and we will add before the statement of the assumption that that is the range considered.
>
> **Reply to** “$P(Q_i \leq 0)$”: This is another good point, and **we can shed some light between theorem** 3.1 **and the fact that** $P(Q_i \leq 0)$ is small. Let $sd(Q_i)$ be the standard deviation of the random variable $Q_i$. By use of *Chebyshev inequality* and the result of theorem 3.1 we have that:
> $P(Q_i \leq 0) \leq P(|Q_i - E[Q_i]| \geq E[Q_i]) \leq sd(Q_i)^2/E[Q_i]^2 \leq C_1^2$ where $C_1$ is introduced in theorem 3.1 and can be made arbitrarily small for large l and small learning rate. From here one can work out directly the bound even if $C_1$, which is made explicit in the appendix, does not have a very nice form. We will add this brief calculation in the main text, as we think it can be informative for the reader.
>
> **Reply to** “$E[Q_i]=0$”: We know that in the stationary phase the expectation of the dot product reported in equation (3) is zero, but the bound that we provide in theorem 3.2 holds for the gradient coherences $Q_i$ (where we cannot assume that the two threads are independent, since they both start from the same point). As $l$ grows one can assume that, for $i > 1$, the two components in $Q_i$ are getting closer to independence and then equation (3) kicks in. But **our proof does not assume independence**.
>
> **Reply to “interpret thm 3.1 and 3.2”:** We will add an explicit interpretation of the results of theorem 3.1 and 3.2 with the aid of figure 3.
>
> **Reply to “compare SplitSGD and other methods”:** Thanks for pointing out these empirical baselines. We will try to compare SplitSGD with these methods in our final version.
>
> **Reply to “step size too small”:** In figure 4 we see that **when the learning rate is very small, the diagnostic is correctly never reducing the learning rate**. This means that the performance of SplitSGD is better than the methods that deterministically reduce the learning rate (Decreasing and Half in the legend) but slightly worse than SGD with constant learning rate, since during the diagnostic the two parallel threads get close to the minimizer slower than a single thread of double length. Reviewer #3 suggested a possible modification to the procedure that allows for the learning rate to be increased if some condition is met. This would certainly give an advantage against SGD with constant learning rate even in the regime of very small learning rate. When the learning rate is larger, the diagnostic is indeed beneficial compared to constant learning rate.
>
> **Reply to “several runs”:** This last point that you raise is excellent, and we are taking care of it now by **running all of the experiments in figure 5 for four additional times**. We will then report in the main paper the average trajectories, and in the appendix the plot with also the confidence bands (as we suspect that for plots that are already pretty dense the confidence bands could make it hard for the reader to distinguish the trajectories). We would like to point out, though, that some of the **key insights that we get from these plots cannot be attributed to the randomness of a single run**, for example when we see that both SGD and Adam severely overfit during the training of the CNN on Fashion-MNIST and LSTM on Penn Treebank, while SplitSGD does not.

---

### Official Review · AnonReviewer1 · 2020-10-29

**Rating:** 7
**Confidence:** 3

**Review:**

The paper introduces SplitSGD method that detects the stationary phase in the stochastic optimization process and shrinks the learning rate. The SplitSGD is based on the observation that before reaching the stationary phase, two random batches of data will likely to have the gradient aligned as the noise between different batches is dominated by the shared gradient, whereas after reaching the stationary phase, two random batches should have misaligned gradient as the gradient has become mainly noise. This observation is intuitive, and some theoretical results show that (a) at the beginning, the algorithm determines non-stationary with high probability, and (b) more important, the SplitSGD algorithm is guaranteed to converge with probability tending to 1. The experiment reveals the advantage of the SplitSGD method over alternative SGD algorithms for CNN, Resnet, and LSTM.

Personally, I feel this paper is good enough to be accepted. The intuition is neat, and the new approach, as supported by both the theoretical and empirical results, should merit significant values to be widely known to other scholars. The paper has made enough contributions and has high clarity in terms of writing.

Here are some concerns that I would suggest the author to consider:
1. The theoretical results are of course important, however, the proven results could appear expected and not have surprise, despite the many technical challenges. The proven results are either for the initial steps, or for the final state on whether the algorithm converges. To further enhance the significance of the theoretical results, it could seem better to establish results on the convergence process, such as the convergence rate analysis that the author in this paper already states would be “left for future work” and the analysis “appears to be a very challenging problem”.
2. The datasets in the empirical evaluation seem not to have large sizes, especially considering the availability of large datasets nowadays. It would make the paper more convincing if the authors can add larger datasets for comparison of methods.
3. The proposed method has gains over a number of alternatives in the simulation. The gap between the new method and the other methods appears not really large. More comparison could be helpful, although I do not think it is fully critical.
4. For the simulation results, as far as I understand, the SplitSGD has better test metrics and results in less overfitting. It would be very helpful if the authors could provide an intuitive explanation of why this is the case. Also, could early stopping achieve a similar performance?
5. Here is a typo: in Eq (5) and in line 21 of algorithm 1, $\theta^{(k)}_{i\cdot l}$ should have $i\cdot l +1$ rather than $i\cdot l$ in the subscript, to match the definition in Eq (4).

---

> ### Author Response · Authors · 2020-11-15
> **Response to Reviewer #1**
>
> Dear reviewer #1, thanks for your insightful comments! In particular, we appreciate your recognition of our efforts in providing both **theoretical and empirical results**. Below our responses to your concerns.
>
> 1. We agree that also getting the convergence rate would be an improvement on the current convergence result. Unfortunately, this has been proven to be very challenging because of the dependence of SplitSGD on the splitting diagnostic, whose output is described in equation (7) but for which it is hard to express the probability of the two events explicitly. In particular, we know that the probability of observing a large number of consecutive diagnostics that report that convergence has not been reached goes to $0$, but a non asymptotic analysis of this convergence appears to be extremely cumbersome.
> 2. Thanks for your suggestions; we will try to experiment with larger datasets. It is worthwhile to notice that the three datasets that we used for deep learning, Fashion-MNIST (more complex than MNIST), CIFAR-10, and Penn Treebank, are **standard benchmarks for deep learning**. Specifically, each of Fashion-MNIST and CIFAR-10 has 60K examples. As for Penn treebank, there are about 1.1M examples (tokens).
> 3. We are currently running **extra simulations for each of the experiments reported in figure 5**, so that we can plot an average trajectory (and also add the confidence bands, possibly in a second plot in the appendix) and get a better sense of how large the gap with the other methods and improve the strength of our experiment section.
> 4. As we see in figure 5, **early stopping is definitely beneficial for SGD, Adam and FDR to avoid overfitting**. It appears that SplitSGD does not suffer much from this problem. We do not have a complete analysis of why this is the case, but our intuition is that a **combined effect of the averaging after the diagnostic and the learning rate decay** is the key. Especially in the experiment with ResNet, we see that the averaging immediately brings the test accuracy up, but that when this is not combined with a learning rate decay in the next epoch we see a quick drop of the performance to the previous level. However, when both these factors combine, we see a positive jump followed by a reduction of the oscillations and a new higher level of test accuracy achieved.
> 5. You are correct, thanks for spotting this typo. We will also incorporate your other suggestions accordingly.

---

### Official Review · AnonReviewer3 · 2020-10-29

**Rating:** 7
**Confidence:** 3

**Review:**

**Summary**:

The paper focuses on estimating when stochastic gradient dynamics have reached a stationary phase by considering the inner product of pairwise stochastic trajectories referred to as threads. The chosen approach avoids strongly correlated estimates which leads to better mixing and more reliable identification of a stationary phase than previous work for certain problems. The proposed algorithm is specifically adapted for Deep Learning problems and performs well on the presented synthetic and real world problems.

**Reason for score**:

A prevalent problem of ML/DL is laid out and solution is provided that is backed by theory, intuition, simulations and empirical experiments and altogether presented in a systematic comprehensible manner. These are all constituents of a great paper that I also think provides a valid contribution to the field.




**Pros**:

- The paper is well-written and makes good use of visual media to convey results and intuition.
- A drawback of previous methods for stationarity detection is the (potentially) correlated samples obtained from successive gradient steps. This did not seem to be a problem in the DL case presented in the experiments but would be for more general problems. The paper presents a clever way to address this problem.
- All decisions regarding parameters and modifications to the algorithm are properly backed up with good arguments, empirical evidence and theory.

**Questions**:

- There is also one question regarding the implementation of SplitSGD for the Deep Learning experiments that I might have missed. Is it the case that each epoch of SplitSGD requires 2 passes through the training set or is it alternating between 1 epoch standard training and 1 epoch diagnostic?

- SplitSGD is generalized to use different optimizers such as SGD with momentum and Adam and I see no reason why other first-order methods could not be incorporated as well. In ML these optimizers often rely on a diagonal preconditioning (call it $P_t$) where the update $\theta_{t+1}= \theta_t - \eta_t P_t g_t$ is used. If we assume $P_t=P$ to be constant during the diagnostic we end up with average gradients of $P\bar{g}_i$ and $Q_i=<P\bar{g}_i^{(1)},P\bar{g}_i^{(2)}>=<\bar{g}_i^{(1)},\bar{g}_i^{(2)}>_{P^2}$. Do you know how this affects the estimates $Q_i$ and whether it would be necessary to invert the preconditioning for such adaptive methods? How was this handled for SplitAdam?

- In theorem 3.1 and 3.2 and by extension figure 3 the effects of large learning rates and long time-horizons for the diagnostic are compared. Let's assume the horizon $t_1$ is fixed to a "suitable" value. It then looks as if a more general adaptation scheme could be constructed from the condition in Eq.7. Denote the sum on the LHS of Eq.7 as $\Sigma$. If the learning rate is too small, the red histogram of Figure 1 is expected -> $\Sigma$ would be small, which for the chosen $t_1$ has not lead to stationarity. Would it then be possible to slightly increase the learning rate to bring the next diagnostic closer to stationarity? An extreme case of which could be to replace lines 8-12 of the algorithm with:
```
if S:
	decrease eta by factor gamma
else:
	increase eta by factor gamma/2
```

Did you consider such an adaptation or see any advantages/disadvantages of such a procedure?



**Tips**:

- In figure 3 you could flip the colors in the left pair of histograms to have "stationary distribution as blue" consistently.
- The x-labels of figure 6 are difficult to read. You could replace such a label with just a pair of parameters in larger size. Ex. "w=10, q=0.35" -> "(10, 0.35)" and explain the ($w$,$q$) setup in the caption.
- Search for "improvement is only mandatory is" and replace last "is" with "as"
- At the end of section 4.3 the values for $w$ and $q$ got mixed up.

---

> ### Author Response · Authors · 2020-11-15
> **Response to Reviewer #3**
>
> Dear reviewer #3, thanks for your insightful comments. We really appreciate the depth of your understanding of our work. We will address below your questions and your tips.
>
> **Questions:**
>
> - In the deep learning experiments, as we also do in the convex setting, we first run single-thread SGD for a fixed number of epochs $t_1$ (which is set to $t_1 = 4$). We then allocate one epoch for the splitting diagnostic, so that **each of the two threads has access to half of the data**. This means that $2\cdot l\cdot w$ is equal to the size of the dataset, and each thread of the diagnostic is updated using $l\cdot w$ data points. Each epoch that we consider for SplitSGD has the same number of points as the ones in SGD, so the extra computational cost just consists of computing the inner products. After averaging the final points of the two threads of the diagnostic we restart single-thread SGD.
> - This is a very good point and it certainly deserves a better understanding of the theory underlying the behavior of the $Q_i$ for different optimizers. What we did is to simply use the right hand side of equation (5) for both SGD with momentum and Adam. This means that, on a window of length $l$, we look at the **difference between the first and last point on each of the two threads**, and then compute the inner product (the rescaling is not crucial since we will then only look at the sign of the $Q_i$).
> - This could be a very good idea and it would certainly help for extreme cases where the initial learning rate is small, for example in the right plots of figure 4. In those cases, we would probably be able to perform better than SGD with constant learning rate for all choices of the initial learning rate. In general we feel that this could be a good idea to **speed up the initial convergence of the method**, mostly for convex problems. Our main concerns with this modification are the following: (i) it adds another hyperparameter to the model, where the increasing factor $\frac{1}{2}$ could be any $\rho \in (0, 1)$ and (ii) it is not immediately clear to us if this could have more severe drawbacks in deep learning, where stationarity might not be detected for a certain number of consecutive diagnostics but increasing the learning rate every time would bring it to be too large for the problem at hand. A possible alternative could be to consider two different thresholds, $q_1$ and $q_2$, with $q_1 < q_2$, one for decaying the learning rate and the other one for increasing it. This of course would not solve problem (i), but might be better overall for the following reason. With this modification condition (7) would become:
>     * decay if $\Sigma > q_2\cdot w$
>     * increase by same factor $\gamma$ if $\Sigma < q_1\cdot w$
>     * keep constant otherwise
> Further study would be needed to verify if this strategy can speed up the initial performance of SplitSGD also in deep learning, and how to tune $q_1$ and $q_2$.
>
> **Tips:**
>
> Those are all excellent tips, we will incorporate them in the final version of the paper.

---

### Author Response · Authors · 2020-11-24
**Revised paper submitted**

Dear Reviewers and Area Chair, we submitted a revised version of our paper, which has been improved according to the many good suggestions that we have received. We have done our best to accommodate as many of the reviewers' comments as possible. In particular:
-  We have improved the explanation of theorem 3.1 by adding a complete calculation to prove that $P(Q_i < 0)$ is small.
-  We have added a lemma in the main text (previously in the appendix) that we think makes the understanding of theorem 3.2 easier.
-  For both theorems 3.1 and 3.2 we explain more in details the connection with figure 3, which has been modified according to suggestions.
-  We have run an extensive simulation to improve the experiment section. Specifically, now each trajectory in figure 5 is an average of 5 runs (and we reported the plots with confidence bands in the appendix). All the claims that we have made in the original paper about the excellent performance of SplitSGD still hold, and we expanded on them in the DNN section.
-  We have added an important reference (Pesme et al. (2020)) in the related work section.
-  We have corrected all the typos that we have been made aware of, and we thank the reviewers that have carefully read our paper for spotting those.

---

### Decision · Program_Chairs · 2021-01-07
**Final Decision**

**Decision:**

Reject

**Comment:**

This paper proposes to automatically determine when the SGD step-size should be decreased, by running two "threads" of SGD for a bunch of iterations, divide those into windows, and then look at the average inner-product of the gradients in the two threads in each window. If the inner-product tends to be high, that indicates that there is still "signal" in the gradient and it should not be decreased. If it is low, that indicates that the gradient is mostly "noise". In the latter case, the learning rate is decreased by a factor of gamma and the length of the next phase is increased by gamma.

Theorem 3.1 essentially assumes smoothness, a bounded fourth moment for the stochastic gradient, and that the stochastic gradient error is not too far from isotropic. Then it shows that if the step-size is set small enough, the standard deviation of the diagnostic (Q_i) can be upper-bounded in terms of the expected value of Q_i. It follows that the probability of Q_i being negative cannot be too large (bounded in terms of the step size eta and the length of the windows l).

Theorem 3.2 adds the assumption of strong convexity and weakens the assumption on the gradient to a bounded second moment. Then it upper-bounds the expected value of the diagnostic in terms of its standard deviation.

Proposition 3.4 gives a proof of convergence. As far as I can tell the proof is essentially that the learning rate decay can't be much worse than what would happen if the diagnostic *always* set to decrease. In particular: (1) It's impossible for the learning rate to decay too quickly, since the length of each phase is increased by gamma whenever the learning rate is decreased by gamma. (This is a "non-adaptive result.) (2) The learning rate will eventually decay with probability 1.

Various concerns were brought up by the reviewers. Perhaps the most strongly voiced concern was that the proposed method is a heuristic rather than a method with a rigorous guarantee. For my part I am in agreement with the authors and other reviewers that heuristic methods for decreasing the learning rate are worthy of study given the large practical importance of this problem.

I concur with the concern raised by some reviewers that the theoretical component of the paper may not have little explanatory value for the results that are given. The assumption of strong convexity is not a major concern to me. (Though not true it can still give intuition.) More concerning is that theory essentially takes a fixed step-size scheme (repeatedly decrease the step size by gamma and increasing the length of a phase by gamma) and then shows that the diagnostic can’t be too much worse. This isn’t in keeping with the motivation of being adaptive.

The reviewers were also concerned about the explanation of better results due to less overfitting. This may be true, but the theory makes no mention of overfitting.

There was a consensus that the experimental results were promising, though some minor issues were raised.

While the direction explored in the paper has value, there are enough open questions about the relationship of the theory to the experimental results to warrant another round of review.

Small thoughts, not significant to acceptance:

The current heuristic runs two separate threads and looks at the inner-product of those gradients. An alternative to this would be to run a single thread along with a "ghost" thread that computes a different gradient at each iteration. It would be great to comment on the difference and why one might be superior to the other. A more radical alternative would be to run a single thread, but then compute the diagnostic on each half of the minibatch. A more radical alternative still would be to analytically do that splitting many times and average the results. This seems like it might simultaneously reduce the variance of the diagnostic and also reduce the computational cost.

2. The current heuristic runs two threads. Is there a tradeoff if you run more?

3. The statement of theorems could be more user-friendly. To understand Thm 3.1, I needed to search o find the definitions of: eta, l, i, w, Q_i. With a small amount of effort this could be re-written to remind the reader that w is the number of windows, l is the length, eta is the stepsize, etc. It is particularly unfortunate that sd() is never formally defined (only by reading the appendix did I discover that this was the standard deviation.)

4. The fact that the length of threads is always increased by a factor of gamma whenever the step size is reduced by gamma seems contrary to the spirit of the proposed diagnostic. After all, this "bakes in" a kind of "fastest possible" decay schedule. If the diagnostic were fully reliable, shouldn't this not be necessary? The decision to add this doe not get nearly enough discussion in the paper in my view.

5. I think it might be clearer to re-state theorem 3.1 including the Chebyshev result after it.